# Quantification of fibrosis extend and airspace availability in lung: A semi-automatic ImageJ/Fiji toolbox

**Bertrand-David Ségard** [1]*, **Kodai Kimura**[1], **Yuimi Matsuoka**[1], **Tomomi Imamura**[1], **Ayana Ikeda**[1], **Takahiro Iwamiya**[1,2]

1 Research and Development Department, Metcela Inc., Kawasaki, Kanagawa, Japan, 2 Institute for Advanced Biosciences, Keio University, Tsuruoka, Yamagata, Japan

* b.segard@metcela.com

**Data Availability Statement:** Macros are available on GitHub: DOI: 10.5281/zenodo.10669353 Repository URL: https://github.com/Metcela-Code/

## Abstract

The evaluation of the structural integrity of mechanically dynamic organs such as lungs is critical for the diagnosis of numerous pathologies and the development of therapies. This task is classically performed by histology experts in a qualitative or semi-quantitative manner. Automatic digital image processing methods appeared in the last decades, and although immensely powerful, tools are highly specialized and lack the versatility required in various experimental designs. Here, a set of scripts for the image processing software ImageJ/Fiji to easily quantify fibrosis extend and alveolar airspace availability in Sirius Red or Masson's trichrome stained samples is presented. The toolbox consists in thirteen modules: sample detection, particles filtration (automatic and manual), border definition, air ducts identification, air ducts walls definition, parenchyma extraction, MT-staining specific pre-processing, fibrosis detection, fibrosis particles filtration, airspace detection, and visualizations (tissue only or tissue and airspace). While the process is largely automated, critical parameters are accessible to the user for increased adaptability. The modularity of the protocol allows for its adjustment to alternative experimental settings. Fibrosis and airspace can be combined as an evaluation of the structural integrity of the organ. All settings and intermediate states are saved to ensure reproducibility. These new analysis scripts allow for a rapid quantification of fibrosis and airspace in a large variety of experimental settings.

## Introduction

Respiratory diseases are one of the leading causes of disability and death worldwide, through a decline in lung function [1–3]. The induction of fibrosis is a major factor in this functional decline as the increase in stiffness of fibrotic tissues leads to mechanical defects. Additionally, the increased volume of these tissues leads to the collapse of alveolar structures and a decrease in gas exchange capacity.

The etiology of lung fibrosis is varied and incompletely described. The main disorders known to promote scarring are idiopathic pulmonary fibrosis (IPF), collagen vascular

Lung-fibrosis-and-airspace-quantification The protocol is available on Protocols.io (publication synchronized with PLOS ONE): RESERVED DOI: 10.17504/protocols.io.14egn7nqmv5d/v1 Repository URL: https://www.protocols.io/view/ quantification-of-fibrosis-extend-and-airspace-ava-b9ztr76n Raw data, settings, results, and figures are available on Figshare: Repository URL: https:// figshare.com/projects/Supporting_data_for_ PLOS_ONE_S_gard_2024_/146790".

**Funding:** All authors of the present article are current or former employees of Metcela Inc. This study was funded by grants from Kawasaki City (https://www.city.kawasaki.jp/en/index.html), namely "Kawasaki City New Technology/New Product Development Support Project Subsidy" (Grant No. 84; awarded to TIw) and "Kawasaki City "New Normal" Research and Development Subsidy" (Grant No. 225; awarded to TIw). Sample preparation, imaging, and method development were financed by these grants. Metcela provided support in the form of salaries for authors BDS, KK, YM, TIm, AI, and TIw. Funders did not have any additional role in the study design, data collection and analysis, decision to publish, or preparation of the manuscript. The specific roles of these authors are articulated in the "author contributions" section.

**Competing interests:** I have read the journal's policy and the authors of this manuscript have the following competing interests to declare. All authors of the present article are current or former employees of Metcela Inc. Metcela is developing cell therapies to treat chronic organ diseases. Metcela's core patented technology involves a particular population of cardiac fibroblasts, namely VCAM-1-positive cardiac fibroblasts. VCAM-1-positive cardiac fibroblasts are known to replenish and re-establish the damaged cardiac muscles and the microenvironment surrounding them. Two products for heart failure patients are currently being tested in phase I clinical trial. Takahiro Iwamiya is a co-founder and co-CEO of Metcela Inc. and has ownership of stocks. TIw has the authority to make payment decisions regarding employee salaries. This does not alter our adherence to PLOS ONE policies on sharing data and materials.

disorders, telomere disease, Erdheim-Chester disease (ECD), and Hermansky-Pudlak syndrome (HPS) [4, 5]. Additionally, the pandemic of COVID-19 leads to a surge of new cases of lung fibrosis [6, 7]. Therefore, there is an urgent need to develop therapies for pulmonary fibrosis [1, 2]. One important milestone in such endeavor is the standardization of the method to evaluate lung structural integrity in experimental models or clinical specimens. Affordability, speed, and flexibility are desirable to accelerate research globally.

The precise quantification of pulmonary fibrosis is challenging, and not standardized [8]. Because of technical challenges, most methods are applied on random fields instead of whole slide scans [5]. Indeed, the most widely used method to estimate lung fibrosis was proposed by Thomas Ashcroft and colleagues in 1988, and consists on averaging discrete scores defined at random positions throughout the sample [9]. While immensely useful, this scoring method requires advanced expertise in histopathology, and present considerable inter- and intra-scorer variability [5]. As the gold standard of pulmonary fibrosis evaluation, refinements of this approach are regularly proposed [8, 10, 11]. More recently, computer assisted methods for quantifying collagen in samples stained with Picrosirius Red (SR) were developed and applied to various organs (e.g., heart [12], liver [13, 14]). These approaches proved robust compared to traditional strategies but still consist in random fields image analysis. A systematic analysis of the whole tissue section is desirable as the distributions of anatomical features and structural defects are not homogeneous throughout the organ [15, 16]. In this regard, an innovative approach based on the morphological classification of collagen fibers according to their compactness was recently proposed [17]. This algorithm proved accurate and applicable to various organs but specializes in Picrosirius Red staining and relies on proprietary software for histological structures segmentation. Methods allowing for the quantification of fibrosis in Masson's trichrome (MT) stained samples were also proposed (e.g., heart [18], lung [19]) but are not applicable to other staining. Thus, the quantification of fibrosis in lung remains an open topic. Moreover, the combination of fibrosis extend and airspace availability is not accessible in current methods. This is however desirable as the study of both tissue and air compartments can account for other phenomena such as compensatory growth.

Here, a set of macros for ImageJ/Fiji allowing to quantify fibrosis and airspace in lung whole slices stained with Sirius Red or Masson's trichrome is presented. This toolbox is built as an ensemble of specialized modules to ensure maximal flexibility and adaptability to various experimental designs. Additionally, numerous parameters can be defined by the user while keeping histological expertise requirements minimal. All critical settings and intermediate states are saved to improve reproducibility. Fibrosis and airspace quantification in SR- and MT-stained mouse lung samples is demonstrated in the present study.

## Materials and methods

The protocol described in this peer-reviewed article is published on protocols.io (DOI: https://dx.doi.org/10.17504/protocols.io.14egn7nqmv5d/v1) and is included for printing purposes as S1 File.

The efficiency of the protocol is demonstrated using a benchmark dataset of samples stained with Sirius Red or Masson's trichrome (Fig 1).

All animal experiments were approved by the Animal Experiment Committee of Innovation Center of NanoMedicine (Kawasaki, Japan). Approval numbers are: SKL-A-20005 and A20-003-1.

Animals were anesthetized with isoflurane. Sacrifice was made by cervical dislocation. All efforts were made to minimize animal suffering.

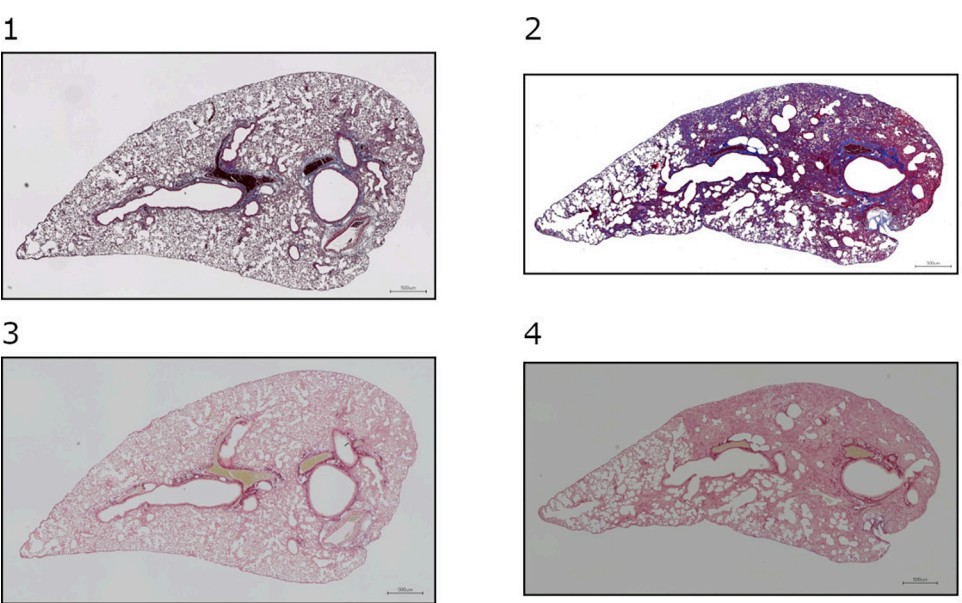

**Fig 1. Representative samples.** 1. Mouse lung slice stained with MT (baseline sample); 2. Mouse lung slice stained with MT (bleomycin-induced fibrosis sample); 3. Mouse lung slice stained with SR (baseline sample); 4. Mouse lung slice stained with SR (bleomycin-induced fibrosis sample).

## Induction of lung fibrosis in mice

The animal model was produced by Jackson Laboratory Japan (Tsukuba, Japan), samples were prepared by GenoStaff (Tokyo, Japan), and subsequent analysis was performed in-house.

Pulmonary fibrosis was induced in C57BL/6J mice (male, 9 weeks old) as previously reported [20]. Briefly, a bleomycin (BLM) solution (1 U/mL in 0.9% saline) was administered by a liquid MicroSprayer (PennCentury; Wyndmoor, PA) at a concentration of 2 mg/kg in 50 μL into the respiratory tract of mice. Animals in the baseline group (N = 5; representing mild fibrosis) were sacrificed 7 days after BLM treatment. Animals in the BLM-induced fibrosis group (N = 5; representing advanced fibrosis) were injected with a saline solution intravenously on day 7 of the experiment. These animals were sacrificed 14 days after injection.

Mouse lung samples were fixed in 4% paraformaldehyde (PFA), embedded in paraffin, and sectioned with a microtome. Lung sections 5 μm in thickness were stained with Sirius Red or Masson's trichrome. Images were captured using an All-in-One Fluorescence Microscope BZ-X710 (Keyence; Osaka, Japan).

## Fibrosis quantification pipeline

In both staining configurations, the fibrosis quantification protocol is performed in the following order: 1) cleaning of the slice, 2) definition of the border, 3) definition of air ducts and air ducts walls, 4) extraction of the parenchyma, 5) quantification of fibrosis in the parenchyma, and 6) assembly of the tissue visualization.

## Airspace quantification pipeline

The airspace is identified within the parenchyma defined in the fibrosis quantification pipeline. Briefly, the parenchyma is virtually closed by the border and air ducts walls, and alveoli are defined as the negative space in the sample.

## Expected results

The toolbox comprises thirteen specialized modules applicable to RGB images. Scale measurement must be performed before analysis. Each module is described hereafter.

### Module 1: Sample detection

This module allows to detect the sample and remove the background. It is based on a double thresholding over the blue channel (Fig 2).

Briefly, the image is scaled and duplicated. The channel of interest is extracted, and a double thresholding is applied. All selected particles are described (i.e., area and perimeter), and a visualization is assembled.

Output consists of: (1) mask of ROI, (2) overlay of ROI over original image, (3) ROI file, (4) measurements (area and perimeter) of all selected particles, (5) summary of measurements (area and perimeter) of all selected particles, and (6) settings (input file name, scale, lower threshold, upper threshold, and number of selected particles).

### Module 2a: Particle filtering (automatic)

This module allows to remove false positive particles from the selection. It is based on a size filter over a binary mask (e.g., output of module 1) (Fig 3).

Briefly, the image is duplicated, and a size filter is applied. All remaining particles are described (i.e., area and perimeter), and a visualization is assembled.

Output consists of: (1) mask of ROI, (2) overlay of ROI over original mask, (3) ROI file, (4) measurements (area and perimeter) of all remaining particles, (5) summary of measurements (area and perimeter) of all remaining particles, and (6) settings (input file name, minimum particle size, maximum particle size, and number of remaining particles).

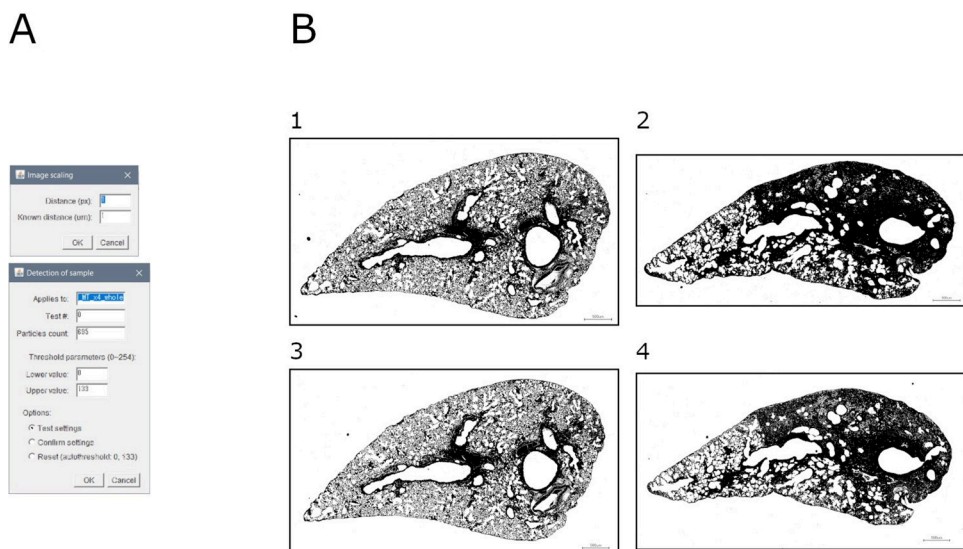

**Fig 2. Sample detection.** A. Interface of the "sample detection" tool. B. Binary masks of detected samples: 1. Mouse lung slice stained with MT (baseline sample); 2. Mouse lung slice stained with MT (bleomycin-induced fibrosis sample); 3. Mouse lung slice stained with SR (baseline sample); 4. Mouse lung slice stained with SR (bleomycin-induced fibrosis sample).

A

B

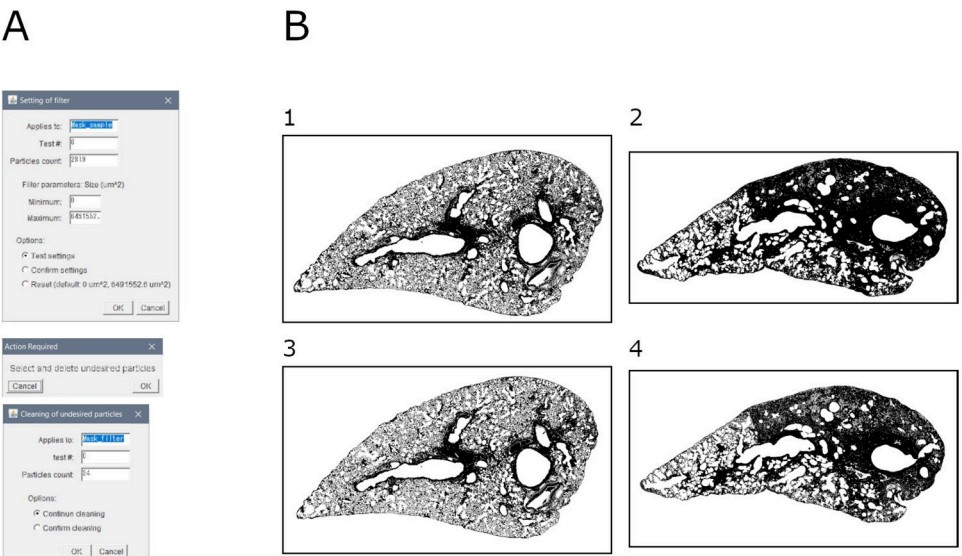

**Fig 3. Background cleaning.** A. Interface of the "particle filter" automatic and manual tools. B. Cleaned binary masks of samples: 1. Mouse lung slice stained with MT (baseline sample); 2. Mouse lung slice stained with MT (bleomycin-induced fibrosis sample); 3. Mouse lung slice stained with SR (baseline sample); 4. Mouse lung slice stained with SR (bleomycin-induced fibrosis sample).

## Module 2b: Particle filtering (manual)

This module allows to remove false positive particles in the background which cannot be filtered by size. It is based on a manual deletion of objects over a binary mask (e.g., output of module 2a) (Fig 3).

Briefly, the image is duplicated, and background particles are deleted by the user. All remaining particles are described (i.e., area and perimeter), and a visualization is assembled.

Output consists of: (1) mask of ROI, (2) overlay of ROI over original mask, (3) ROI file, (4) measurements (area and perimeter) of all remaining particles, (5) summary of measurements (area and perimeter) of all remaining particles, and (6) settings (input file name and number of remaining particles).

## Module 3: Border definition

This module allows to remove the pleura from the whole sample. It is based on a reduction of the sample ROI at a defined distance over a binary mask (e.g., output of module 2b) (Fig 4).

Briefly, the image is duplicated, and the user is invited to repair potential breaks in the sample. Then, the pleura thickness is set by a defined distance from the border of the sample. Finally, a visualization is assembled.

Output consists of: (1) mask of ROI, (2) overlay of ROI over original mask, (3) ROI file, and (4) settings (input file name and pleura thickness).

## Module 4: Air ducts identification

This module allows to define the air ducts. It is based on a manual identification of anatomical structures over a binary mask (e.g., output of module 2b) (Fig 5).

Briefly, the image is duplicated, and anatomical structures are identified by the user. A visualization is assembled in real-time.

Output consists of: (1) overlay of ROI over original mask, and (2) ROI file.

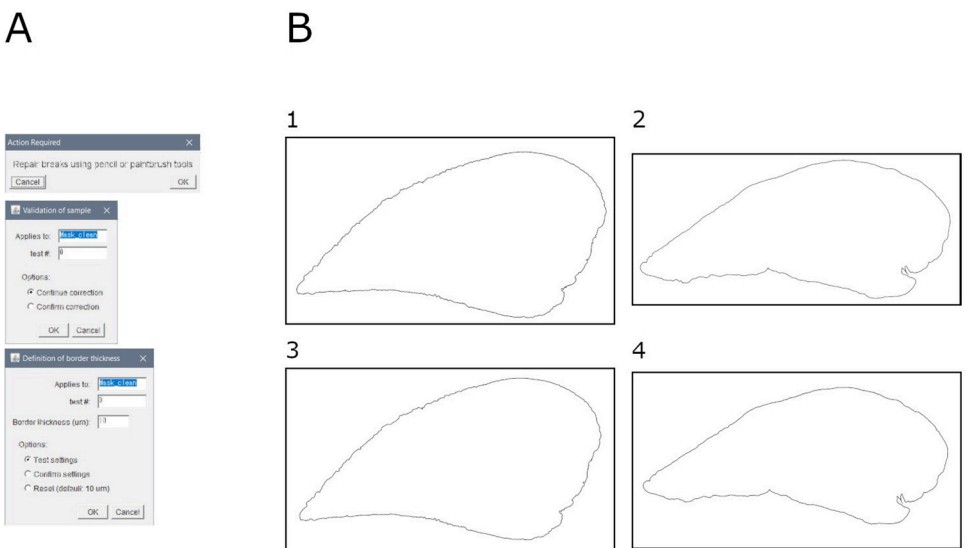

**Fig 4. Border definition.** A. Interface of the "border definition" tool. B. Binary masks of the borders of the samples: 1. Mouse lung slice stained with MT (baseline sample); 2. Mouse lung slice stained with MT (bleomycin-induced fibrosis sample); 3. Mouse lung slice stained with SR (baseline sample); 4. Mouse lung slice stained with SR (bleomycin-induced fibrosis sample).

## Module 5: Air ducts walls definition

This module allows to remove the physiological depositions of collagen around air ducts (i.e., peri-duct collagen deposition). It is based on an extension of the air ducts ROI (i.e., output of module 4) at a defined distance over a binary mask (e.g., output of module 2b) (Fig 6).

Briefly, the image is duplicated, and the air ducts walls are set at a defined distance around the air ducts. A visualization is assembled.

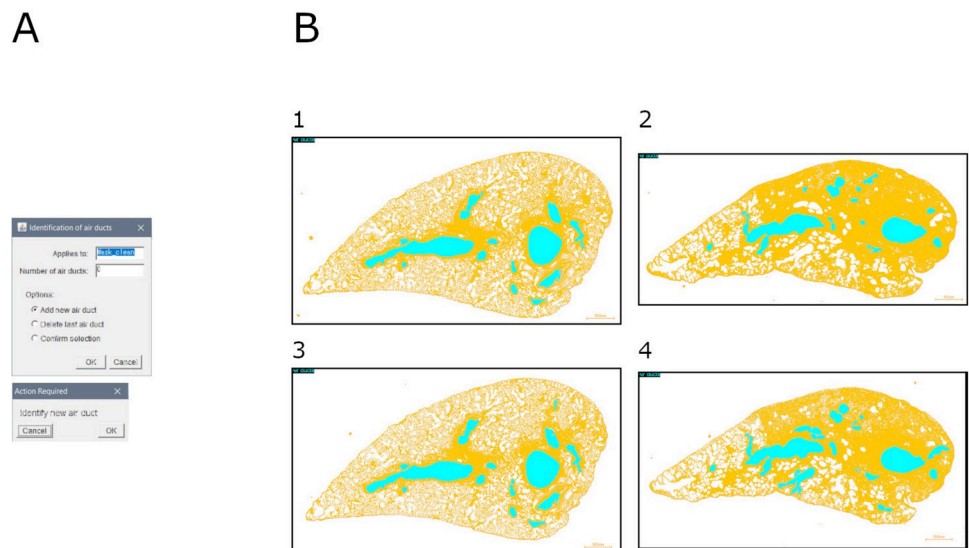

**Fig 5. Air ducts identification.** A. Interface of the "air duct identification" tool. B. Overlays of samples and identified air ducts: 1. Mouse lung slice stained with MT (baseline sample); 2. Mouse lung slice stained with MT (bleomycin-induced fibrosis sample); 3. Mouse lung slice stained with SR (baseline sample); 4. Mouse lung slice stained with SR (bleomycin-induced fibrosis sample).

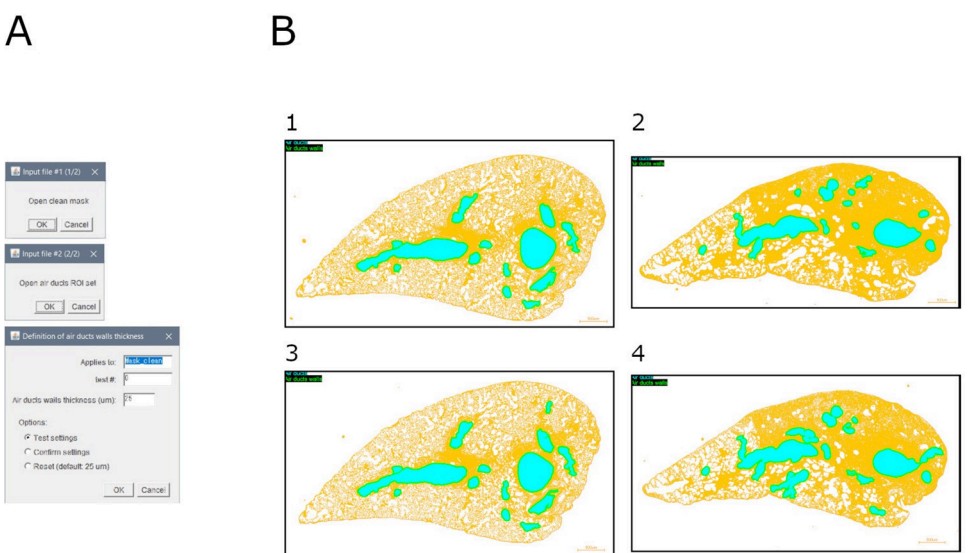

**Fig 6. Air ducts walls definition.** A. Interface of the "air ducts walls definition" tool. B. Overlays of samples, identified air ducts, and defined air ducts walls: 1. Mouse lung slice stained with MT (baseline sample); 2. Mouse lung slice stained with MT (bleomycin-induced fibrosis sample); 3. Mouse lung slice stained with SR (baseline sample); 4. Mouse lung slice stained with SR (bleomycin-induced fibrosis sample).

Output consists of: (1) overlay #1 (visualization of air ducts walls as an extended ROI), (2) overlay #2 (visualization of air ducts walls and air ducts), (3) ROI file, and (4) settings (input file name and air ducts walls thickness).

## Module 6a: Parenchyma extraction

This module allows to extract the parenchyma from the original image. It is based on the deletion of the ROI defined in the previous steps (i.e., background, pleura, and air ducts walls) (Fig 7).

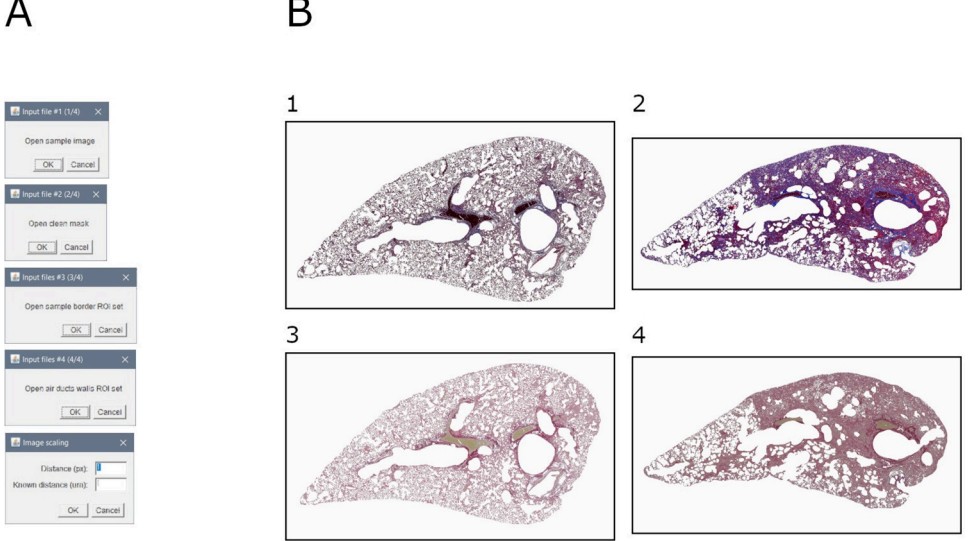

**Fig 7. Parenchyma extraction.** A. Interface of the "parenchyma extraction" tool. B. Images of parenchyma of samples: 1. Mouse lung slice stained with MT (baseline sample); 2. Mouse lung slice stained with MT (bleomycin-induced fibrosis sample); 3. Mouse lung slice stained with SR (baseline sample); 4. Mouse lung slice stained with SR (bleomycin-induced fibrosis sample).

Briefly, the original image is duplicated, and undesirable regions are deleted.

Output consists of: (1) RGB image of the parenchyma, (2) measurements (area and perimeter) of the parenchyma, (3) ROI file, and (4) settings (input file name and scale).

### Module 6b: MT pre-processing

This module allows to separate the connective tissue, the cytoplasm, and the nuclei in the image of the parenchyma. It is based on the color deconvolution of Masson's trichrome staining.

Briefly, the image is duplicated, and the "Colour Deconvolution" plugin (https://imagej. net/plugins/colour-deconvolution; pre-installed in Fiji) is applied.

Output consists of: (1) RGB image of the connective tissue in the parenchyma, (2) RGB image of the cytoplasm in the parenchyma, and (3) RGB image of the nuclei in the parenchyma.

### Module 7: Fibrosis detection

This module allows to detect the pathological depositions of collagen. It is based on a double thresholding over the green channel (e.g., output of modules 6a/b) (Fig 8).

Briefly, the image is scaled and duplicated. The channel of interest is extracted, and a double thresholding is applied. All selected particles are described (i.e., area and perimeter), and a visualization is assembled.

Output consists of: (1) mask of ROI, (2) overlay of ROI over the parenchyma image, (3) ROI file, (4) measurements (area and perimeter) of all selected particles, (5) summary of measurements (area and perimeter) of all selected particles, and (6) settings (input file name, scale, lower threshold, upper threshold, and number of selected particles).

### Module 8: Fibrosis particle filtering (automatic)

This module allows to remove false positive particles from the fibrosis selection. It is based on a size filter over a binary mask (e.g., output of module 7) (Fig 9).

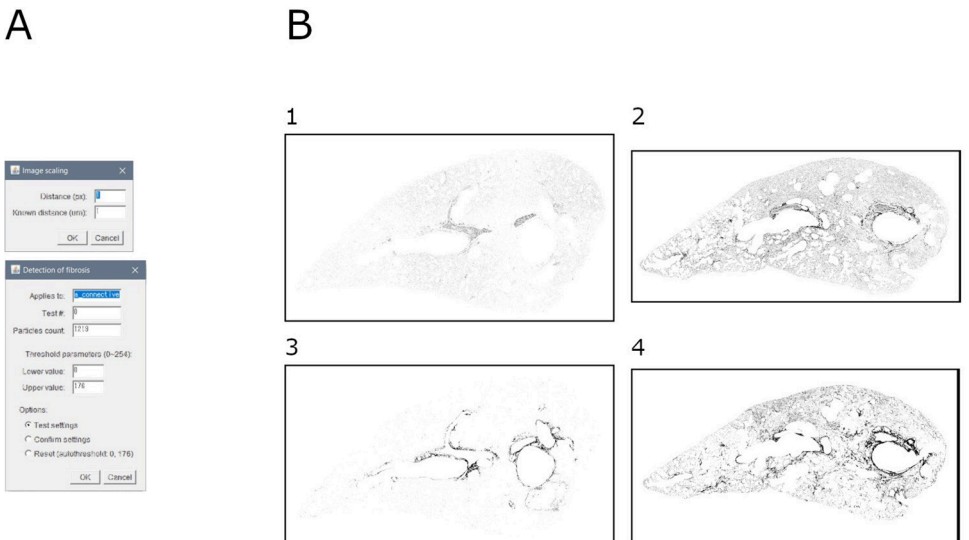

**Fig 8. Fibrosis detection.** A. Interface of the "fibrosis detection" tool. B. Binary masks of fibrosis in samples: 1. Mouse lung slice stained with MT (baseline sample); 2. Mouse lung slice stained with MT (bleomycin-induced fibrosis sample); 3. Mouse lung slice stained with SR (baseline sample); 4. Mouse lung slice stained with SR (bleomycin-induced fibrosis sample).

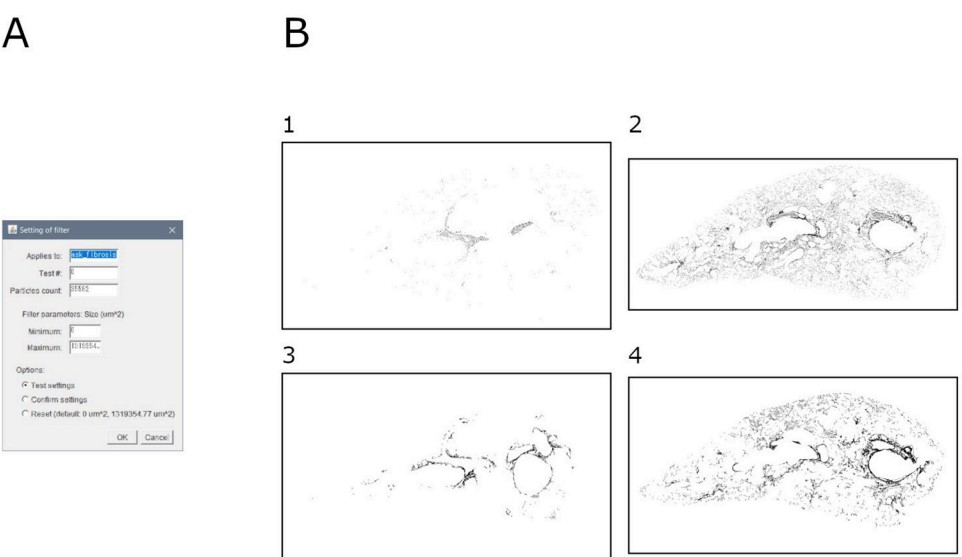

**Fig 9. Fibrosis cleaning.** A. Interface of the "fibrosis filter" automatic tool. B. Binary masks of filtered fibrosis in samples: 1. Mouse lung slice stained with MT (baseline sample); 2. Mouse lung slice stained with MT (bleomycin-induced fibrosis sample); 3. Mouse lung slice stained with SR (baseline sample); 4. Mouse lung slice stained with SR (bleomycin-induced fibrosis sample).

Briefly, the image is duplicated, and a size filter is applied. All remaining particles are described (i.e., area and perimeter), and a visualization is assembled.

Output consists of: (1) mask of ROI, (2) overlay of ROI over original mask, (3) ROI file, (4) measurements (area and perimeter) of all remaining particles, (5) summary of measurements (area and perimeter) of all remaining particles, and (6) settings (input file name, minimum particle size, maximum particle size, and number of remaining particles).

## Module 9: Visualization of tissue

This module allows to assemble a visualization of the ROI defined in previous steps (i.e., parenchyma, border, air ducts, air ducts walls, and fibrosis) over the original image (Fig 10).

Briefly, the original image is duplicated, and the various regions of interest are highlighted.

Output consists of: (1) RGB image of the sample with ROI highlighted.

## Module 10: Airspace detection

This module allows to detect the airspace in the parenchyma of the sample. It can be applied irrespective of the staining used (Fig 11).

Briefly, the parenchyma is duplicated, and virtually closed by the pleura and air ducts walls previously defined. The airspace is defined as the negative space in the parenchyma. Particles are filtered by size and circularity.

Output consists of: (1) mask of ROI, (2) overlay of ROI over original image, (3) ROI file, (4) measurements (area and perimeter) of all selected particles, (5) summary of measurements (area and perimeter) of all selected particles, and (6) settings (input file name, minimum particle size, maximum particle size, minimum particle circularity, maximum particle circularity, and number of selected particles).

A

B

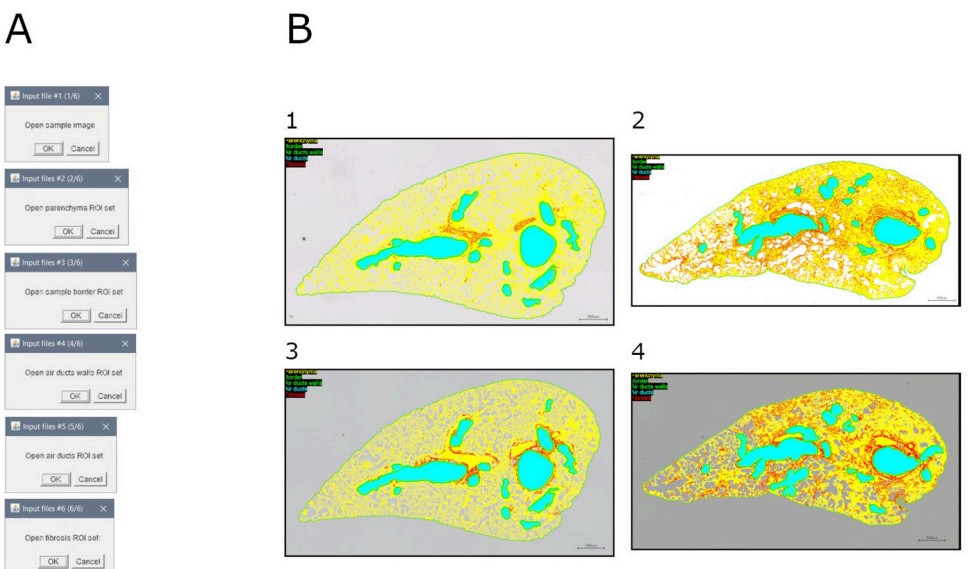

**Fig 10. Tissue visualization.** A. Interface of the "visualization tissue" tool. B. Visualization of regions of interest in samples: 1. Mouse lung slice stained with MT (baseline sample); 2. Mouse lung slice stained with MT (bleomycin-induced fibrosis sample); 3. Mouse lung slice stained with SR (baseline sample); 4. Mouse lung slice stained with SR (bleomycin-induced fibrosis sample).

## Module 11: Visualization of tissue and airspace

This module allows to assemble a visualization of the ROI defined in previous steps (i.e., parenchyma, border, air ducts, air ducts walls, fibrosis, and airspace) over the original image (Fig 12).

Briefly, the visualization of tissue image (i.e., output of module 9) is duplicated, and the airspace is highlighted.

A

B

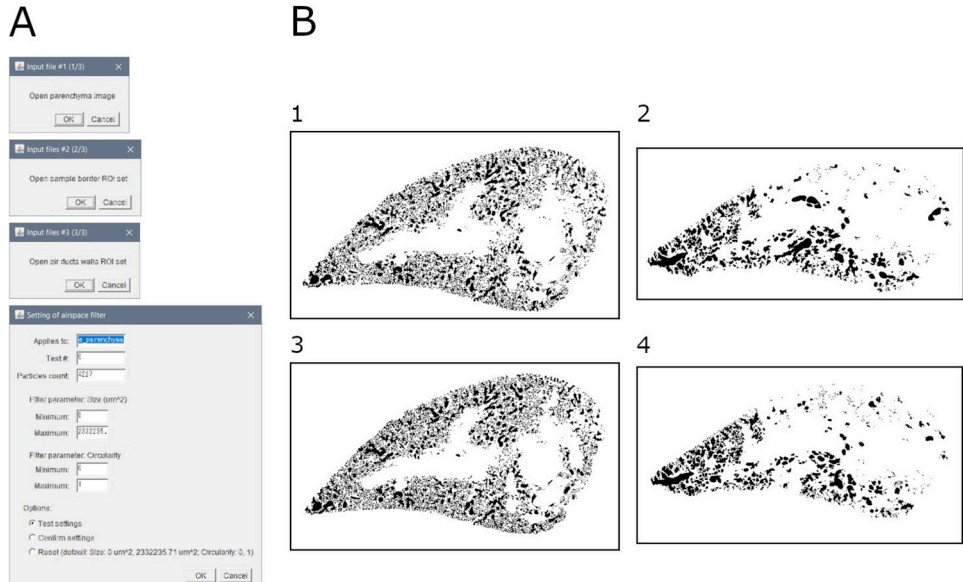

**Fig 11. Airspace quantification.** A. Interface of the "airspace detection" tool. B. Binary masks of airspace in samples: 1. Mouse lung slice stained with MT (baseline sample); 2. Mouse lung slice stained with MT (bleomycin-induced fibrosis sample); 3. Mouse lung slice stained with SR (baseline sample); 4. Mouse lung slice stained with SR (bleomycin-induced fibrosis sample).

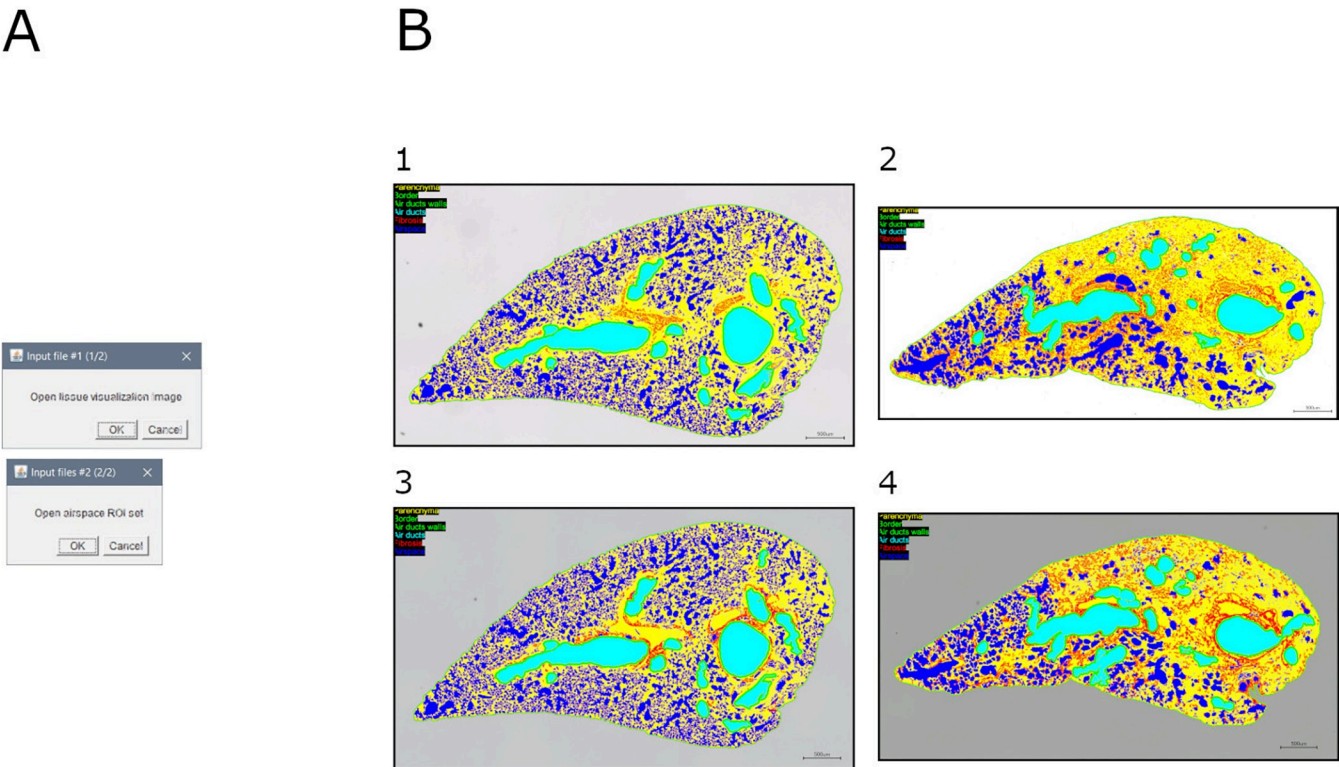

**Fig 12. Tissue and airspace visualization.** A. Interface of the "visualization tissue airspace" tool. B. Visualization of regions of interest in samples: 1. Mouse lung slice stained with MT (baseline sample); 2. Mouse lung slice stained with MT (bleomycin-induced fibrosis sample); 3. Mouse lung slice stained with SR (baseline sample); 4. Mouse lung slice stained with SR (bleomycin-induced fibrosis sample).

Output consists of: (1) RGB image of the sample with ROI highlighted.

## Compilation of results

Critical results are: (1) parenchyma area (output of module 6a), (2) fibrosis area (output of module 8), and (3) airspace area (output of module 10) (Figs 13 and 14).

## Interpretation of results

Fibrosis is a major complication in numerous pathologies. Scaring induces a rigidification of tissues which leads to compromised mechanical properties of dynamic organs such as lung or

| Staining | Group | Image | Parenchyma area (um^2) | Fibrosis area (um^2) | Ratio fibrosis/parenchyma | Ratio fibrosis/parenchyma (average) | Ratio fibrosis/parenchyma (SD) | Airspace area (um^2) | Ratio airspace/sample | Ratio airspace/sample (average) | Ratio airspace/sample (SD) |
|---|---|---|---|---|---|---|---|---|---|---|---|
| MT | baseline | No.001_MT_x4_whole | 5200759.00 | 45867.12 | 0.88% | | | 2322982.88 | 30.88% | | |
| | | No.002_MT_x4_whole | 5144331.00 | 8761.60 | 0.17% | | | 4060025.38 | 44.11% | | |
| | | No.003_MT_x4_whole | 5529444.50 | 11834.32 | 0.21% | 0.74% | 0.51% | 1587579.44 | 22.31% | 32.96% | 7.35% |
| | | No.004_MT_x4_whole | 5943846.50 | 51359.67 | 0.86% | | | 2557957.12 | 30.09% | | |
| | | No.005_MT_x4_whole | 5314531.50 | 82091.46 | 1.54% | | | 3180409.447 | 37.44% | | |
| | bleomycin | No.316_MT_x4_whole | 5918923.00 | 883752.30 | 14.93% | | | 1749946.20 | 22.82% | | |
| | | No.317_MT_x4_whole | 7289913.50 | 633747.90 | 8.69% | | | 1609855.55 | 18.09% | | |
| | | No.318_MT_x4_whole | 6263939.00 | 453717.40 | 7.24% | 10.84% | 3.81% | 2724016.55 | 30.31% | 22.59% | 6.36% |
| | | No.322_MT_x4_whole | 6920177.50 | 1104203.20 | 15.96% | | | 1054583.06 | 13.22% | | |
| | | No.324_MT_x4_whole | 5019126.00 | 370452.55 | 7.38% | | | 2000995.33 | 28.50% | | |
| SR | baseline | No.001_SR_x4_whole | 4220880.00 | 79653.37 | 1.89% | | | 3239348.192 | 43.42% | | |
| | | No.002_SR_x4_whole | 4237791.50 | 73261.38 | 1.73% | | | 4589556.398 | 51.99% | | |
| | | No.003_SR_x4_whole | 4988987.50 | 165278.69 | 3.31% | 2.93% | 1.01% | 2287315.547 | 31.44% | 40.30% | 7.39% |
| | | No.004_SR_x4_whole | 5888834.00 | 194668.16 | 3.31% | | | 2959707.609 | 33.45% | | |
| | | No.005_SR_x4_whole | 4961241.50 | 219624.34 | 4.43% | | | 3475217.337 | 41.19% | | |
| | bleomycin | No.316_SR_x4_whole | 5310043.50 | 981253.20 | 18.48% | | | 1764021.294 | 24.94% | | |
| | | No.317_SR_x4_whole | 5959201.00 | 968400.73 | 16.25% | | | 1886163.158 | 24.04% | | |
| | | No.318_SR_x4_whole | 5691023.00 | 433552.88 | 7.62% | 12.78% | 4.69% | 3555701.524 | 38.45% | 28.21% | 7.59% |
| | | No.322_SR_x4_whole | 6379155.00 | 937650.67 | 14.70% | | | 1411343.054 | 18.12% | | |
| | | No.324_SR_x4_whole | 4446453.5 | 304085.396 | 6.84% | | | 2445704.569 | 35.49% | | |

**Fig 13. Results (data).** Quantification of fibrosis extend and airspace availability in the benchmark dataset (all samples).

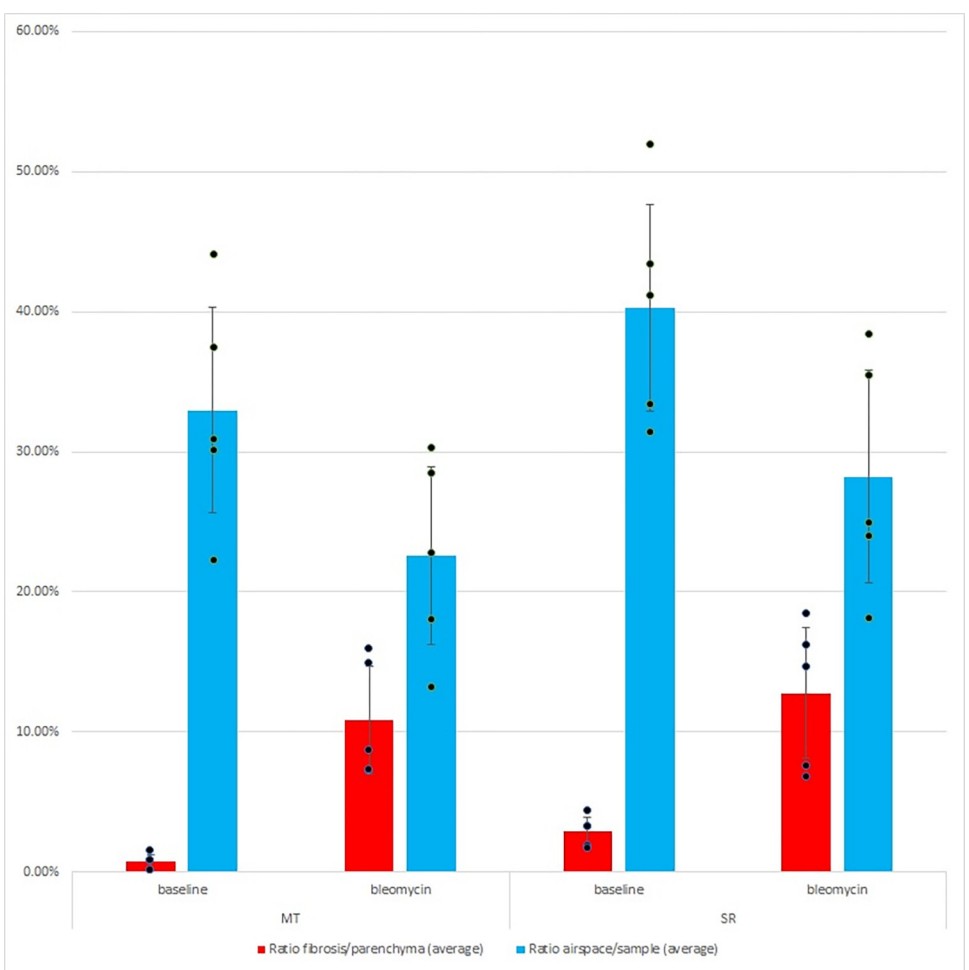

**Fig 14. Results (graph).** Quantification of fibrosis extend and airspace availability in the benchmark dataset (all samples).

heart. The development of therapies requires the quantification of fibrosis extend and the evaluation of the structural integrity of the organ. While numerous staining methods are available, fibrosis is often evaluated manually by histology experts. Indeed, the Ashcroft scoring is still the gold standard for pulmonary fibrosis evaluation. However, this approach presents considerable variability between scorers due to the subjectivity of scores assignment [5]. Additionally, it is based on a discrete scale consisting of only nine digits. Moreover, this method cannot be applied to whole organ slices [9]. More recently, computer assisted quantification methods were developed with amazing success but, these approaches are limited to specific experimental settings.

The fibrosis quantification method presented here can be applied to samples stained with SR or MT. The most important results obtained are the area of the parenchyma and the area of collagen fibers (in SR) or dense connective tissue (in MT). The fibrosis extend can thus be expressed as a fraction of the parenchyma area. This allows for the objective comparison of samples intra- and inter-experiments.

As neither SR nor MT distinguishes cell types, the exclusion of cellular infiltrates (e.g., blood clots) is challenging. This can lead to errors in the evaluation of the remodeling of tissues. For example, cellular infiltrates are observed within the lung parenchyma in HPS

pulmonary fibrosis, and alveolar macrophages can be retained in bronchoalveolar area [5]. Similarly, the accumulation of mucus cannot be discriminated with SR staining alone [5]. As the volume of air exchanged in lungs is directly correlated to survival, the evaluation of airspace in samples is desirable. The approach presented here allows to measure geometrical characteristics of alveoli and is not affected by the type of alveolus blockage. Additionally, the area of airspace can be combined with the area of parenchyma to calculate the tissue density. While it cannot substitute for functional tests such as forced vital capacity (FVC), it complements the histological study of lung tissues.

While commercial solutions to quantify a variety of features in micrographs are available, they are only partially satisfying in an academic context as they act as expensive black boxes. Non-commercial tools aiming at quantifying pulmonary fibrosis are based on the classification of micro-tiles as tissue or not-tissue throughout the sample, which results in a low-resolution mapping of the slice [5, 11, 19, 21]. Although some aspects of the present protocol remain subjective (e.g., threshold values, filtering cutoff, air ducts identification), it is fast and reliable. It outputs reproducible results at maximal resolution (i.e., no-tiling of image). Only a general knowledge in anatomy and microscopy is required, relieving research teams from the dependency on histology experts. Moreover, its modularity allows for a smooth adaptation to various experimental settings. Finally, the free and open-source nature of the toolbox makes it easily accessible to a large variety of research facilities and authorizes improvements by the community.

The present method is expected to improve the consistency of sample comparisons within and between studies, accelerate the speed of therapy development for COVID-19-derived pulmonary fibrosis, and bridge the gap in lung functionality between clinical and animal studies.

## Supporting information

**S1 Fig.**
(TIF)

**S2 Fig.**
(TIF)

**S3 Fig.**
(TIF)

**S4 Fig.**
(TIF)

**S1 File. Step-by-step protocol, also available on protocols.io.**
(PDF)

**S2 File.**
(PDF)

**S3 File.**
(ZIP)

## Acknowledgments

The authors gratefully thanks Mr. Abhiraj Kesharwani (University of Tokyo, Japan) for reviewing this manuscript.

## Author Contributions

**Conceptualization:** Bertrand-David Ségard, Takahiro Iwamiya.

**Data curation:** Bertrand-David Ségard.

**Formal analysis:** Bertrand-David Ségard.

**Funding acquisition:** Takahiro Iwamiya.

**Investigation:** Kodai Kimura, Tomomi Imamura, Ayana Ikeda.

**Methodology:** Bertrand-David Ségard, Yuimi Matsuoka, Takahiro Iwamiya.

**Project administration:** Takahiro Iwamiya.

**Resources:** Takahiro Iwamiya.

**Software:** Bertrand-David Ségard.

**Supervision:** Takahiro Iwamiya.

**Validation:** Bertrand-David Ségard.

**Visualization:** Bertrand-David Ségard.

**Writing – original draft:** Bertrand-David Ségard.

**Writing – review & editing:** Bertrand-David Ségard, Yuimi Matsuoka, Takahiro Iwamiya.

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
