## [Decision Letter · Decision Letter 0]

8 Sep 2023

PONE-D-23-22513Quantification of fibrosis extend and airspace availability in lung: a semi-automatic ImageJ/Fiji toolboxPLOS ONE

Dear Dr. Ségard,

Thank you for submitting your manuscript to PLOS ONE. After careful consideration, we feel that it has merit but does not fully meet PLOS ONE’s publication criteria as it currently stands. Therefore, we invite you to submit a revised version of the manuscript that addresses the points raised during the review process.

We look forward to receiving your revised manuscript.

Kind regards,

Panayiotis Maghsoudlou

Academic Editor

PLOS ONE

Journal Requirements:

3. Thank you for stating the following in the Competing Interests: 

   "I have read the journal's policy and the authors of this manuscript have the following competing interests:

Samples preparation, imaging, and method development were financed by Metcela Inc. in the form of salaries for BDS, KK, YM, TIm, AI, and TIw.

Takahiro Iwamiya is a co-founder and co-CEO of Metcela Inc. and has ownership of stocks. Other authors are employees of Metcela Inc. Takahiro Iwamiya has the authority to make payment decisions regarding employee salaries."

We note that one or more of the authors have an affiliation to the commercial funders of this research study : Metcela Inc. 

6. Please ensure that you refer to Figures 1-11 in your text as, if accepted, production will need this reference to link the reader to the figure.

7. Please include a separate caption for each figure in your manuscript.

8. We note you have not yet provided a protocols.io PDF version of your protocol and/or a protocols.io DOI. When you submit your revision, please provide a PDF version of your protocol as generated by protocols.io (the file will have the protocols.io logo in the upper right corner of the first page) as a Supporting Information file. The filename should be S1_file.pdf, and you should enter “S1 File” into the Description field. Any additional protocols should be numbered S2, S3, and so on. Please also follow the instructions for Supporting Information captions [https://journals.plos.org/plosone/s/supporting-information#loc-captions]. The title in the caption should read: “Step-by-step protocol, also available on protocols.io.”

Please assign your protocol a protocols.io DOI, if you have not already done so, and include the following line in the Materials and Methods section of your manuscript: “The protocol described in this peer-reviewed article is published on protocols.io (https://dx.doi.org/10.17504/protocols.io.[...]) and is included for printing purposes as S1 File.” You should also supply the DOI in the Protocols.io DOI field of the submission form when you submit your revision.

If you have not yet uploaded your protocol to protocols.io, you are invited to use the platform’s protocol entry service [https://www.protocols.io/we-enter-protocols] for doing so, at no charge. Through this service, the team at protocols.io will enter your protocol for you and format it in a way that takes advantage of the platform’s features. When submitting your protocol to the protocol entry service please include the customer code PLOS2022 in the Note field and indicate that your protocol is associated with a PLOS ONE Lab Protocol Submission. You should also include the title and manuscript number of your PLOS ONE submission.

Reviewers' comments:

Reviewer's Responses to Questions

**Comments to the Author**

1. Does the manuscript report a protocol which is of utility to the research community and adds value to the published literature?

Reviewer #1: Yes

2. Has the protocol been described in sufficient detail?

To answer this question, please click the link to protocols.io in the Materials and Methods section of the manuscript (if a link has been provided) or consult the step-by-step protocol in the Supporting Information files.

The step-by-step protocol should contain sufficient detail for another researcher to be able to reproduce all experiments and analyses.

Reviewer #1: Yes

3. Does the protocol describe a validated method?

Reviewer #1: No

4. If the manuscript contains new data, have the authors made this data fully available?

Reviewer #1: No

**5. Is the article presented in an intelligible fashion and written in standard English?**

Reviewer #1: Yes

6. Review Comments to the Author

Reviewer #1: The submitted manuscript (protocol) by Ségard et al. is well written and every step of the protocol is understandable and easy to follow. The rationale behind this work is clear and the developed set of freely available macros certainly is of utility for the research community concerned with lung diseases.

Comments:

Review Question 3: Does the protocol describe a validated method? Answer: No

While the authors made an effort to validate this algorithm in an appropriate animal model for fibrosis, certain information is missing to definitively speak about a validated method.

1) Is it correct that only n=1 sample per condition (MT Baseline, MT Bleo, SR Baseline, SR Bleo) was used for validation? If so - in the reviewer's opinion - this is not sufficient for validation. (If a larger dataset is available, then it was not available to the reviewer as the figshare link is not yet activated/working)

 The reason for "Review Question 4: If the manuscript contains new data, have the authors made this data fully available?" being answered is no, is that if there is a bigger dataset and it is not available, it seems to not be fully available yet.

2) In the materials and methods section, the description of the bleomycin-induced pulmonary fibrosis in mice is a bit confusing. Did animals in the baseline group receive saline via the MicroSprayer? Why were they sacrificed on day 7, but the bleomycin mice were sacrificed after 21 days. What is the use of the intravenous saline injection in these mice? How many animals per group were used?

 If indeed mice were compared at different time points, this would methodologically not be ideal

3) Have the authors tried different file formats? Whole slide images can be too large for Fiji/ImageJ. For instance, a single 2 GB file (compressed) of a lung cannot simply be opened in Fiji without strong downsampling of the image (this is the reviewer's experience). Recommendations on how to work with different file formats might be useful for potential users

4) While it is certainly true, especially in the case of diseases with a heterogeneous distribution of the pathology within the organ, that whole slide images are preferred over random fields, this does not solve all the issues with histological analyses of the lungs. It might be useful to advise the reader that ideally, several slides of different anatomical positions within the lungs should be analysed. It might also be useful to refer the reader to literature regarding lung tissue sampling/orientation (for instance how to perform a systematic uniform random sampling, or an isotropic uniform random orientation of the lungs) to reduce sampling or orientation bias.

7. PLOS authors have the option to publish the peer review history of their article (what does this mean?). If published, this will include your full peer review and any attached files.

Reviewer #1: No

---

## [Author Response · Author response to Decision Letter 0]

9 Nov 2023

Journal Requirements:

 and 

The manuscript and associated files were revised to meet PLOS ONE’s style requirements as described in the relevant templates.

Funding information was removed from the manuscript. An updated statement is appended to the cover letter.

3. Thank you for stating the following in the Competing Interests: 

 "I have read the journal's policy and the authors of this manuscript have the following competing interests:

Samples preparation, imaging, and method development were financed by Metcela Inc. in the form of salaries for BDS, KK, YM, TIm, AI, and TIw.

Takahiro Iwamiya is a co-founder and co-CEO of Metcela Inc. and has ownership of stocks. Other authors are employees of Metcela Inc. Takahiro Iwamiya has the authority to make payment decisions regarding employee salaries."

We note that one or more of the authors have an affiliation to the commercial funders of this research study : Metcela Inc.

Funding Statement

All authors of the present article are current or former employees of Metcela Inc.

This study was funded by grants from Kawasaki City (https://www.city.kawasaki.jp/en/index.html), namely “Kawasaki City New Technology/New Product Development Support Project Subsidy” (Grant No. 84; awarded to TIw) and “Kawasaki City “New Normal” Research and Development Subsidy” (Grant No. 225; awarded to TIw). Sample preparation, imaging, and method development were financed by these grants.

Metcela provided support in the form of salaries for authors BDS, KK, YM, TIm, AI, and Tiw. Funders did not have any additional role in the study design, data collection and analysis, decision to publish, or preparation of the manuscript. The specific roles of these authors are articulated in the “author contributions” section.

Competing Interests Statement

I have read the journal's policy and the authors of this manuscript have the following competing interests to declare.

All authors of the present article are current or former employees of Metcela Inc.

Metcela is developing cell therapies to treat chronic organ diseases. Metcela’s core patented technology involves a particular population of cardiac fibroblasts, namely VCAM-1-positive cardiac fibroblast (VCF; patent granted in Japan (No.6241893), Europe and China, and is currently pending in the U.S.). VCAM-1-positive cardiac fibroblasts are known to replenish and re-establish the damaged cardiac muscles and the microenvironment surrounding them. Two products for heart failure patients are currently being tested in phase I clinical trial.

Takahiro Iwamiya is a co-founder and co-CEO of Metcela Inc. and has ownership of stocks. TIw has the authority to make payment decisions regarding employee salaries.

This does not alter our adherence to PLOS ONE policies on sharing data and materials.

Both updated statements are appended to the cover letter.

Data are available on three repositories: GitHub (code; copy joined to the submission), protocols.io (step-by-step protocol; copy joined to the submission), and Figshare (raw data, settings, intermediate files, and figures).

Figshare restricts the access to private projects to named invitation and no general link can be created for this review. Additionally, the complete dataset is too heavy to be joined to the submission.

Samples and settings used as benchmark are joined to this submission to allow replication. Note that slight differences in results may appear due to variations at user-dependent steps (i.e., background cleaning and air duct identification).

All repositories are private now but will be made public at acceptance. Accession numbers and DOI will be generated at this time. I acknowledge that should the manuscript be accepted for publication, it will be held until full access to data is granted.

The position of the ethics statements was corrected as instructed.

6. Please ensure that you refer to Figures 1-11 in your text as, if accepted, production will need this reference to link the reader to the figure.

References to figures were added to the text.

7. Please include a separate caption for each figure in your manuscript.

Figure captions were added at the end of the manuscript.

8. We note you have not yet provided a protocols.io PDF version of your protocol and/or a protocols.io DOI. When you submit your revision, please provide a PDF version of your protocol as generated by protocols.io (the file will have the protocols.io logo in the upper right corner of the first page) as a Supporting Information file. The filename should be S1_file.pdf, and you should enter “S1 File” into the Description field. Any additional protocols should be numbered S2, S3, and so on. Please also follow the instructions for Supporting Information captions [https://journals.plos.org/plosone/s/supporting-information#loc-captions]. The title in the caption should read: “Step-by-step protocol, also available on protocols.io.”

The file “S1_file.pdf” joined to this submission is the PDF version of the document on protocols.io (updated on October 23, 2023).

The protocol was revised for guidelines, figures captions, and results.

Captions in the manuscript were updated following instructions.

Please assign your protocol a protocols.io DOI, if you have not already done so, and include the following line in the Materials and Methods section of your manuscript: “The protocol described in this peer-reviewed article is published on protocols.io (https://dx.doi.org/10.17504/protocols.io.[...]) and is included for printing purposes as S1 File.” You should also supply the DOI in the Protocols.io DOI field of the submission form when you submit your revision.

The link to the private repository is: https://www.protocols.io/private/7D78FD46DC9511EC883C0A58A9FEAC02

The DOI will be generated when the repository is made public.

Reviewers' comments:

Reviewer's Responses to Questions

1. Does the manuscript report a protocol which is of utility to the research community and adds value to the published literature?

Reviewer #1: Yes

2. Has the protocol been described in sufficient detail?

To answer this question, please click the link to protocols.io in the Materials and Methods section of the manuscript (if a link has been provided) or consult the step-by-step protocol in the Supporting Information files.

The step-by-step protocol should contain sufficient detail for another researcher to be able to reproduce all experiments and analyses.

Reviewer #1: Yes

3. Does the protocol describe a validated method?

Reviewer #1: No

4. If the manuscript contains new data, have the authors made this data fully available?

Reviewer #1: No

5. Is the article presented in an intelligible fashion and written in standard English?

Reviewer #1: Yes

6. Review Comments to the Author

Reviewer #1: The submitted manuscript (protocol) by Ségard et al. is well written and every step of the protocol is understandable and easy to follow. The rationale behind this work is clear and the developed set of freely available macros certainly is of utility for the research community concerned with lung diseases.

Comments to the Author

Review Question 3: Does the protocol describe a validated method? Answer: No

While the authors made an effort to validate this algorithm in an appropriate animal model for fibrosis, certain information is missing to definitively speak about a validated method.

1) Is it correct that only n=1 sample per condition (MT Baseline, MT Bleo, SR Baseline, SR Bleo) was used for validation? If so - in the reviewer's opinion - this is not sufficient for validation. (If a larger dataset is available, then it was not available to the reviewer as the figshare link is not yet activated/working).

Thank you for your comment. Samples used to illustrate this protocol were extracted from a larger dataset. These specific samples were selected as they present the best quality of preparation (low background, no breaks) and a large difference in terms of fibrosis progression (“baseline” corresponding to mild fibrosis and “bleomycin” to advanced fibrosis).

In this revision, additional samples of the same groups were added to the demonstration (N = 5 samples per condition). The result table was updated accordingly, and a graph was added (see: protocols.io, step 12). Raw data, settings, intermediate files, and figures are available on Figshare (upon invitation).

 The reason for "Review Question 4: If the manuscript contains new data, have the authors made this data fully available?" being answered is no, is that if there is a bigger dataset and it is not available, it seems to not be fully available yet.

All repositories (i.e., GitHub, protocols.io, and Figshare) will be made public at acceptance. Copies of GitHub and protocols.io data are joined to this submission. However, microscopy data are too heavy to be shared in the same way.

Figshare does not allow to access private repositories by other means than a named invitation. To allow replication, original images and settings are joined to this revision. Note that slight differences in results may appear due to variations at user-dependent steps (i.e., background cleaning and air duct identification).

2) In the materials and methods section, the description of the bleomycin-induced pulmonary fibrosis in mice is a bit confusing. Did animals in the baseline group receive saline via the MicroSprayer? Why were they sacrificed on day 7, but the bleomycin mice were sacrificed after 21 days. What is the use of the intravenous saline injection in these mice? How many animals per group were used?

Thank you very much for your remarks concerning the animal experiment. The original manuscript contained an error as animals in the baseline group were exposed to bleomycin and sacrificed 7 days later. The corresponding paragraph was corrected as follows (lines 125-133 of the revised manuscript):

“Pulmonary fibrosis was induced in C57BL/6J mice (male, 9 weeks old) as previously reported.[20] Briefly, a bleomycin (BLM) solution (1 U/mL in 0.9% saline) was administered by a liquid MicroSprayer (PennCentury; Wyndmoor, PA) at a concentration of 2 mg/kg in 50 µL into the respiratory tract of mice. Animals in the baseline group (N = 5; representing mild fibrosis) were sacrificed 7 days after BLM treatment. Animals in the BLM-induced fibrosis group (N = 5; representing advanced fibrosis) were injected with a saline solution intravenously on day 7 of the experiment. These animals were sacrificed 14 days after injection.”

Samples presented in this study were extracted from a larger dataset. These groups (i.e., bleomycin/baseline D7 and bleomycin/saline D21) were selected as they present the largest difference in terms of fibrosis progression. Indeed, the baseline group can be associated with mild fibrosis, while the bleomycin group presents advanced fibrosis.

The complete dataset cannot be described here, but saline is used as a vehicle control.

 If indeed mice were compared at different time points, this would methodologically not be ideal

Your remark is true, and I hope the correction of the animal experiment description will clarify this point.

These two groups allow to compare mild and advanced fibrosis. Moreover, they allow to test the sensitivity of the protocol in low and high signal intensity samples.

3) Have the authors tried different file formats? Whole slide images can be too large for Fiji/ImageJ. For instance, a single 2 GB file (compressed) of a lung cannot simply be opened in Fiji without strong downsampling of the image (this is the reviewer's experience). Recommendations on how to work with different file formats might be useful for potential users.

The toolbox presented here comprises macros. As such, it is subjected to all limitations of ImageJ/Fiji. While the protocol was not tested on very large images, it should be possible to complete it on any type of image handled by ImageJ/Fiji.

Recommendations on how to work with large images were added to the “before starting” statement on protocols.io as follows:

“ImageJ/Fiji manages images directly in the RAM of the computer. The maximum RAM ImageJ/Fiji is allowed to use is set in a dedicated menu (Edit > Options > Memory & Threads…).

Alternatively, ImageJ/Fiji can import large images as "Virtual Stack" to avoid loading all data in the RAM. A dedicated menu is available to use this format (File > Import > TIFF Virtual Stack…). Note that displaying the image will be slower.

The plugin Bio-Formats supports BigTiff (File > Import > Bio-Formats) and can manage extremely large images.”

4) While it is certainly true, especially in the case of diseases with a heterogeneous distribution of the pathology within the organ, that whole slide images are preferred over random fields, this does not solve all the issues with histological analyses of the lungs. It might be useful to advise the reader that ideally, several slides of different anatomical positions within the lungs should be analysed. It might also be useful to refer the reader to literature regarding lung tissue sampling/orientation (for instance how to perform a systematic uniform random sampling, or an isotropic uniform random orientation of the lungs) to reduce sampling or orientation bias.

Thank you very much for pointing out this important observation. Indeed, pathological but also physiological lung structures have heterogeneous distributions within the organ. Thus, the histological analysis of the lung requires specific sampling.

A reference was added to the manuscript to guide the reader toward lung sampling best practices. Both “systematic uniform random sampling” and “isotropic uniform random sampling” are precisely described (sections 4.2.1 and 4.2.3 respectively).

Please see lines 84-87 of the revised manuscript (“A systematic analysis of the whole tissue section is desirable as the distributions of anatomical features and structural defects are not homogeneous throughout the organ.”) and reference Hsia, 2012 (An Official Research Policy Statement of the American Thoracic Society/European Respiratory Society: Standards for Quantitative Assessment of Lung Structure).

Additionally, recommendations on sample preparation were added to the “before starting” statement on protocols.io as follows:

“Recommendations on lung sample preparation can be found in:

Hsia CC, Hyde DM, Ochs M, Weibel ER; ATS/ERS Joint Task Force on Quantitative Assessment of Lung Structure. An official research policy statement of the American Thoracic Society/European Respiratory Society: standards for quantitative assessment of lung structure. Am J Respir Crit Care Med. 2010 Feb 15;181(4):394-418. doi: 10.1164/rccm.200809-1522ST. PMID: 20130146; PMCID: PMC5455840.

Slides of different anatomical positions within the lungs should be analyzed to minimize sampling or orientation bias.”

7. PLOS authors have the option to publish the peer review history of their article (what does this mean?). If published, this will include your full peer review and any attached files.

Do you want your identity to be public for this peer review? For information about this choice, including consent withdrawal, please see our Privacy Policy.

Reviewer #1: No

---

## [Editor Report · Decision Letter 1]

17 Jan 2024

Quantification of fibrosis extend and airspace availability in lung: a semi-automatic ImageJ/Fiji toolbox

PONE-D-23-22513R1

Dear Dr. Ségard,

We’re pleased to inform you that your manuscript has been judged scientifically suitable for publication and will be formally accepted for publication once it meets all outstanding technical requirements.

Kind regards,

Panayiotis Maghsoudlou

Academic Editor

PLOS ONE

---

## [Editor Report · Acceptance letter]

19 Feb 2024

PONE-D-23-22513R1 

PLOS ONE

Dear Dr. Ségard, 

I'm pleased to inform you that your manuscript has been deemed suitable for publication in PLOS ONE. Congratulations! Your manuscript is now being handed over to our production team.

Kind regards, 

on behalf of

Dr. Panayiotis Maghsoudlou 

Academic Editor

PLOS ONE